# Diagnostic Potential of microRNAs in Extracellular Vesicles Derived from Bronchoalveolar Lavage Fluid for Pneumonia—A Preliminary Report

**DOI:** 10.3390/cells11192961

**Published:** 2022-09-22

**Authors:** Yinfang Sun, Ying Xian, Zhiqin Duan, Zhiping Wan, Jianwei Li, Yao Liao, Xiaogang Bi, Zhongdao Wu, Lifu Wang, Kouxing Zhang

**Affiliations:** 1Department of General Intensive Care Unit, The Third Affiliated Hospital of Sun Yat-sen University, Guangzhou 510700, China; 2Comprehensive Ward, West Hospital District, The Second Affiliated Hospital of Guangzhou Medical University, Guangzhou 510000, China; 3Department of Infectious Diseases, The Third Affiliated Hospital of Sun Yat-sen University, Guangzhou 510630, China; 4Guangdong Provincial Key Laboratory of Liver Disease Research, The Third Affiliated Hospital, Sun Yat-sen University, Guangzhou 510630, China; 5Department of Critical Care Medicine, Zhongshan People’s Hospital of Guangdong Province, Zhongshan 528400, China; 6Department of Parasitology, Zhongshan School of Medicine, Sun Yat-sen University, Guangzhou 510080, China; 7Key Laboratory of Tropical Disease Control, Ministry of Education, Sun Yat-sen University, Guangzhou 510080, China; 8Provincial Engineering Technology Research Center for Biological Vector Control, Guangzhou 510080, China; 9KingMed School of Laboratory Medicine, Guangzhou Medical University, Guangzhou 510180, China; 10Guangzhou Key Laboratory for Clinical Rapid Diagnosis and Early Warning of Infectious Diseases, Guangzhou 510180, China

**Keywords:** pneumonia, extracellular vesicles, miR–17–5p, miR–193a–5p, diagnostic biomarkers

## Abstract

Current clinical needs require the development and use of rapid and effective diagnostic indicators to accelerate the identification of pneumonia and the process of microbiological diagnosis. MicroRNAs (miRNAs) in extracellular vesicles (EVs) have become attractive candidates for novel biomarkers to evaluate the presence and progress of many diseases. We assessed their performance as biomarkers of pneumonia. Patients were divided into the pneumonia group (with pneumonia) and the control group (without pneumonia). We identified and compared two upregulated miRNAs in EVs derived from bronchoalveolar lavage fluid (BALF-EVs) between the two groups (*P*_miR–17–5p_ = 0.009; *P*_miR–193a–5p_ = 0.031). Interestingly, in cell-debris pellets and EVs-free supernatants derived from bronchoalveolar lavage fluid (BALF-cell-debris pellets and BALF-EVs-free supernatants), total plasma, and EVs derived from plasma (plasma-EVs), the expression of miR–17–5p and miR–193a–5p showed no difference between pneumonia group and control group. In vitro experiments revealed that miR–17–5p and miR–193a–5p were strikingly upregulated in EVs derived from macrophages stimulated by lipopolysaccharide. MiR–17–5p (area under the curve, AUC: 0.753) and miR–193a–5p (AUC: 0.692) in BALF-EVs are not inferior to procalcitonin (AUC: 0.685) in the diagnosis of pneumonia. Furthermore, miR–17–5p and miR–193a–5p in BALF-EVs had a significantly higher specificity compared to procalcitonin and could be served as a potential diagnostic marker. MiR–17–5p and miR–193a–5p in EVs may be involved in lung inflammation by influencing the forkhead box O (FoxO) signaling pathway and protein processing in endoplasmic reticulum. This study is one of the few studies which focused on the potential diagnostic role of miRNAs in BALF-EVs for pneumonia and the possibility to use them as new biomarkers for a rapid and early diagnosis.

## 1. Introduction

According to the World Health Organization, lower respiratory tract infections (LRTIs) are the fourth leading cause of death globally, causing 2.6 million deaths worldwide in 2019 [1]. In low-income countries, LRTIs are the most common cause of death; meanwhile, pneumonia is the only infectious disease among the top 10 causes of death in high-income economies [1]. Due to their critical condition, immunodeficiency, and requirement of some invasive medical means such as tracheal intubation, most patients in the intensive care unit (ICU) are prone to pulmonary infection [2,3]. Ventilator-associated pneumonia (VAP) is the most common type of ICU-acquired infection, with an incidence of 9–40% in patients with tracheal intubation and the total mortality of VAP is 22–60% [4]. In the ICU, the 30-day mortality rate of patients with moderate-to-severe community-acquired pneumonia (CAP) is as high as 23–47% [5]. Critical patients with pneumonia who received effective antibiotic therapy within four hours were less likely to die than those who received delayed treatment [6].

Currently, the diagnosis of pneumonia is based on suggestive clinical features such as fever, dyspnea, cough, and leukocytosis, supplemented by evidence of lung consolidation on chest X-rays or computed tomography (CT) to make the final diagnosis [7]. However, abnormal clinical manifestations and pulmonary imaging are delayed than infection and failed to identify subclinical disease states [8,9]. Therefore, multiple biomarkers are used to assist in the early identification of pulmonary infection, C-reactive protein (CRP) and procalcitonin (PCT) are the most commonly used [10]. CRP is an acute phase reactant synthesized predominantly in the liver, both infectious and non-infectious insults could elevate levels of serum CRP [11]. Additionally, CRP is also affected by the concomitant tumor, age, and lipid level [12]. Therefore, the low specificity of CRP limits its application value as a biomarker for early specific identification of pulmonary infection [13]. PCT, secreted predominantly by parafollicular cells of the thyroid, is a recognized marker of bacterial infection. However, PCT is also inadequate in the diagnosis of pneumonia. Serum PCT levels do not increase significantly in patients with central nervous system injury, even in the presence of a definite and uncontrolled pulmonary infection [14]. A prospective study showed that the diagnostic accuracy of procalcitonin for pneumonia was low in emergency department patients with dyspnea, and the clinical utility of procalcitonin was lower than expected [15]. However, early identification of pulmonary infection, carrying out pathogenic microbiological diagnosis, and timely and effective anti-infection measures are of great significance to improve the prognosis of patients.

Bronchoalveolar lavage fluid (BALF) includes alveolar biochemical components and cells and the information gained from BALF is regarded to be a complement to lung biopsy pathology [16]. In addition, lavage samples a much larger area of the lungs than can be obtained by the small tissue fragments of transbronchial biopsy or by open biopsy specimens, therefore giving a more representative picture of inflammatory and immunologic changes [17]. Extracellular vesicles (EVs), as an important component of BALF, selectively package a variety of bioactive substances and mediate material transfer and information exchange between cells [18]. EVs derived from BALF (BALF-EVs) are secreted by various cell types including epithelial cells, alveolar macrophages, and endothelial cells, wherein alveolar macrophages and epithelial cells are the main sources [19]. Among the various active components in EVs, microRNAs (miRNAs), the small non-coding RNAs which have the capability of regulating gene expression by promoting their target messenger RNA (mRNA) degradation or inhibiting the translation of target genes [20,21], are considered as the key component for their function and have attracted much attention as a new tool for medical decision-making [22,23].

Limited reports suggest that miRNAs in EVs play an important role in the occurrence and development of pneumonia and are expected to become ideal markers for the diagnosis and monitoring of inflammatory processes [24,25]. However, existing evidence comes mostly from in vitro experiments and animal studies, and there is a lack of research on clinical samples which are closer to the real state of the disease. In particular, the value of endogenous miRNAs in BALF-EVs in the diagnosis and treatment of pneumonia is still less studied. Therefore, in this study, we evaluated the expression of key miRNAs in EVs derived from clinical samples (bronchoalveolar lavage fluid and plasma), so as to explore new specific biomarkers to assist in the diagnosis of pulmonary infection.

## 2. Materials and Methods

### 2.1. Patients and Samples

This study enrolled 74 patients on invasive mechanical ventilation who were admitted to the ICU of The Third Affiliated Hospital of Sun Yat-sen University between October 2020 and October 2021. These patients were divided into two groups: the pneumonia group (patients with pulmonary infection) and the control group (patients without pulmonary infection). The diagnosis of pulmonary infection was made on the basis of a composite reference standard, which included all microbiological tests and clinical adjudication, referring to the diagnostic criteria of CAP [26] and hospital-acquired pneumonia (HAP) [8]. For this study, we excluded the most severely immunocompromised patients—those with acquired immune deficiency syndrome (AIDS), acute leukemias, lymphomas, or other severe congenital immunodeficiency syndromes, those receiving chemotherapy, especially with neutropenia, and those who recently received solid organ or bone marrow transplants [5]. This study was approved by the Ethics Committee of The Third Affiliated Hospital of Sun Yat-sen University (approval no. [2020] 02-254-02), and all patients signed a written informed consent form before specimen collection.

The participants lay supine on the examination bed, and local anesthesia was performed by injecting 1 mL~2 mL 2% lidocaine into the selected pulmonary and subpulmonary segments through the biopsy hole. Then, the sterilized saline stored at 37 °C or room temperature were quickly injected through the operation channel, a total amount of 60 mL~120 mL, and injected several times (20 mL~50 mL each time). Immediately after the injection of normal saline, the bronchoalveolar lavage fluid (BALF) was obtained by suction with appropriate negative pressure, and 15 mL bronchoalveolar lavage fluid was collected (to avoid oral contamination) in sterile sputum collection tubes. Additionally, 10 mL of blood were collected from all individuals and transferred to EDTA-containing tubes to isolate plasma.

### 2.2. Cell Culture and Treatment

Human THP-1 monocytes [25] and human lung pulmonary epithelial cell line A549 [27] were maintained in RPMI1640 and DMEM medium, respectively, containing 10% heat-inactivated fetal bovine serum (FBS), 1% penicillin, and streptomycin at 37 °C, and 5% CO_2_. To reduce the interference of EVs derived from bovine serum, EVs-depleted FBS was used instead. THP-1 cells were cultured in maintenance media and supplemented with 100 ng/mL phorbol 12-myristate 13-acetate (PMA) for 24 h to differentiate them into macrophages (THP-1 derived macrophages [tMACs]) [28]. For lipopolysaccharide (LPS) treatment, cultured tMACs and A549 were incubated with LPS at a final concentration of 1 μg/mL for the indicated time. The control group was treated with an equal volume of phosphate-buffered saline (PBS).

### 2.3. Purification and Identification of Extracellular Vesicles

Extraction of extracellular vesicles was performed by ultracentrifugation. Briefly, the BALF, plasma, and cell culture supernatant were centrifuged at a low speed (300× *g* for 10 min at 4 °C) (15 mL polypropylene tube, swinging bucket rotor, model A-4-44, 5804R Refrigerated Centrifuge, Eppendorf, Hamburg, Germany) to remove floating cells. The removed cell debris pellets were collected as liquid biopsy elements. The resulting supernatants were transferred into 1.5 mL polypropylene tubes (Eppendorf, Hamburg, Germany) with a micropipette, and then centrifuged at 12,000× *g* for 30 min at 4 °C (Fixed angle rotor, angle of 45°, model #3331, D-37520 Refrigerated Centrifuge, Thermo Electron Corporation, Waltham, MA, USA) to dislodge bacteria. The resulting supernatants were transferred into Quick-Seal Centrifuge tubes (Beckman Coulter, Brea, CA, USA) and centrifuged at 100,000× *g* for 90 min at 4 °C in an Optima L-100xp tabletop ultracentrifuge (Swinging bucket rotor, model SW40 Ti, Optima L-100xp, Beckman Coulter, Brea, CA, USA). The resultant pellet (EVs), which were resuspended with phosphate-buffered saline (PBS) and EVs-free supernatants, was stored at −80 °C. Further, negative-staining transmission electron microscopy (TEM) was used to analyze the EVs. The EVs were loaded on a copper grid and negatively stained with 3% (*w*/*v*) aqueous phosphotungstic acid for 1 min. The grid was then examined using an FEI Tecnai G2 Sprit Twin TEM (FEI, Hillsboro, OR, USA). Thereafter, EVs were analyzed by nanoparticle tracking analysis (NTA). The NanoSight NS300 (Malvern, Malvern, UK) instrument equipped with an sCMOS camera, 488 nm laser (Blue), NTA 3.3 Dev Build 3.3.301 software (Version 3.3, Malvern, Malvern, UK), and 749 frames were used. In addition, the markers for EVs (CD63 and CD81) were confirmed by flow cytometry [29].

### 2.4. Candidate miRNAs Selection

Using the combination of keywords and MeSH terms for “pneumonia” and “miRNAs”, we searched PubMed for articles that describe associations between miRNAs and pneumonia. Each article was reviewed, and associated miRNAs (“miRNAs cluster 1”) were recorded. To focus on the miRNAs with a high likelihood of relevance, we considered only miRNAs that had been studied in a population sample to be potential candidates for investigation. For additional details, see Appendix A.

### 2.5. RNA Extraction and Quantitative Reverse Transcription PCR

Total RNA was harvested using TRIzol (Thermo, Waltham, MA, USA), according to the manufacturer’s instructions [30]. The expression of target miRNA was determined using the SYBR Green Master Mix kit (Takara, Maebashi, Japan) [22]. U6 snRNA was used as an internal control, and the fold change was calculated using the 2^−ΔΔCT^ method. For additional details, see Appendix A.

### 2.6. Statistical Analysis

The continuous variables are expressed as mean ± SD values (normally distributed), or as medians [interquartile ranges] (non-normally distributed) and were analyzed using the Student’s *t*-test or Mann–Whitney U test, respectively. The categorical variables are presented as sample rates (constituent ratio); they were compared using the Chi-squared test or Fisher’s exact test. Multiple comparisons among three or more groups were analyzed using a one-way ANOVA test or a Kruskal–Wallis test (non-parametric). Receiver-operating characteristic (ROC) curves were plotted to investigate the diagnostic value of selected miRNAs. The area under the curve (AUC) was calculated to evaluate the performance of these miRNAs in predicting pulmonary infections.

## 3. Results

### 3.1. Patient Characteristics

The baseline characteristics of the patients were similar in the two groups (Table 1). No significant differences in gender or age, and most comorbidities were present between the groups. The incidence of craniocerebral trauma was significantly higher in the control group than in the pneumonia group. This difference was observed because patients in the control group were mainly those who required mechanical ventilation on account of other medical conditions, such as surgery, but without pulmonary infection. The inflammation indicator, procalcitonin (PCT), was significantly higher in the pneumonia group than in the control group. However, the high-sensitivity C-reactive protein (hsCRP) did not differ between both groups.

### 3.2. Characterization of EVs Derived from Different Samples

Transmission electron microscopy (TEM, FEI, Hillsboro, OR, USA) was used to assess the morphology of extracellular vesicles. As shown in Figure 1A, EVs derived from BALF (BALF-EVs), plasma (plasma-EVs) and cell culture supernatant (cell-EVs) have the characteristic cup-shaped morphology. The size distribution profile of three kinds of EVs was investigated by nanoparticle tracking analysis (NTA), which revealed a peak size of 97 nm, 63 nm, and 118 nm, respectively (Figure 1B). Furthermore, the markers for EVs, CD63, and CD81, in all vesicles were revealed by flow cytometry (Figure 1C).

### 3.3. Differentially Expressed miRNAs Specifically Showed in Extracellular Vesicles Derived from Bronchoalveolar Lavage Fluid

Eighteen miRNAs (miR–542–3p, miR–16–5p, miR–20a–3p, miR–27a–5p, miR–92a–3p, miR–342–3p, miR–422a, miR–423–5p, miR–582–3p, miR–885–5p, miR–193b–5p, miR–432–5p, miR–493–3p, miR–452–5p, miR–200b–3p, miR–34a–3p, miR–17–5p, and miR–193a–5p), which were previously revealed to have dysregulated expression profiles in inflammatory diseases using RNA sequencing of population sample [7,31,32,33,34,35], were selected. Among them, 16 miRNAs between the two groups showed no statistical difference (Appendix A). MiR–17–5p and miR–193a–5p levels were remarkably upregulated in BALF-EVs of the pneumonia group, compared with the control group (Figure 2A). Thereafter, the expression of these two miRNAs in cell-debris pellets derived from BALF (BALF-cell-debris pellets, Figure 2B) and EVs-free supernatants derived from BALF (BALF-EVs-free supernatants, Figure 2C) were assessed. However, no significant differences were observed between both groups. Notably, the expression of miR–17–5p in BALF-EVs-free supernatants was too low to be detected.

Additionally, the levels of these two miRNAs in total plasma, plasma-EVs, and EVs-free supernatants derived from plasma (plasma-EVs-free supernatants) were measured. Surprisingly, in plasma-EVs (Figure 2D), both miR–17–5p and miR–193a–5p showed no difference in level between the two groups; the same trend was observed for total plasma (Figure 2E). Additionally, the expression of miRNAs in plasma-EVs-free supernatants was too low to detect.

### 3.4. Inflammatory Stimuli Increases the Expression of miR–17–5p and miR–193a–5p in Extracellular Vesicles Derived from Macrophages

Due to the presence of several interfering factors in clinical samples, in vitro experiments were conducted to verify that the inflammatory stimuli increase the expression of miR–17–5p and miR–193a–5p in extracellular vesicles. Both miR–17–5p and miR–193a–5p could not be detected in EVs from human alveolar epithelial cells (A549) due to extremely low expression levels. Compared with the control group, miR–17–5p and miR–193a–5p were strikingly upregulated in EVs from tMACs (tMACs-EVs) of the LPS group (Figure 3A). The same trend was not observed for tMACs (Figure 3B) and EVs-free supernatants derived from tMACs (tMACs-EVs-free supernatants, Figure 3C). These results suggest that the expression levels of miR–17–5p and miR–193a–5p in EVs have the potential to suppose whether there is an inflammatory response.

### 3.5. The Expression of miR–17–5p and miR–193a–5p in Extracellular Vesicles Derived from Macrophages Is Dynamic in Inflammatory Response

To explore the relationship between the expression of miR–17–5p and miR–193a–5p in tMAC-EVs and inflammatory response, the levels of these two miRNAs in tMAC-EVs were dynamically measured. In tMACs-EVs (Figure 4A), the expression of miR–193a–5p was drastically increased in a time-dependent manner after LPS treatment and peaked at 12 h; it gradually returned to normal levels. Concurrently, miR–17–5p showed the same trend. In tMACs and tMAC-EVs-free supernatants, the levels of the miR–17–5p and miR–193a–5p were unaffected by LPS stimuli (Figure 4B,C). Further prospective studies are needed to validate whether the expression of miR–17–5p and miR–193a–5p in BALF-EVs of patients with pneumonia have the same variation pattern.

### 3.6. Diagnostic Value of miR–17–5p and miR–193a–5p in BALF-EVs for Pneumonia

The analysis of the usefulness of hsCRP, PCT, miR–17–5p, and miR–193a–5p in BALF-EVs for the diagnosis of pneumonia used receiver-operating-characteristic (ROC) curve techniques. As shown in Table 2, hsCRP had no diagnostic utility. PCT, miR–17–5p and miR–193a–5p in BALF-EVs had acceptable diagnostic value. For miR–17–5p, the area under the curve (AUC) was 0.753, and the sensitivity and specificity values were 59.02 and 84.62%. For the miR–193a–5p cutoff value of 2.40, the AUC was 0.692 and the sensitivity and specificity were 50.82 and 100%. The best AUC for PCT was 0.685, with 62.30% sensitivity and 69.23% specificity using optimal cutoff values. In addition, we performed a logistic regression analysis to combine these two candidate miRNAs, as it is shown in Table 2, the diagnostic utility did not improve by combining miR–17–5p and miR–193a–5p.

### 3.7. MicroRNAs Target Prediction and Pathway Analysis

Targets for miR–17–5p and miR–193a–5p were obtained with TargetScan (http://www.targetscan.org/ accessed on 3 November 2021), miRDB (http://mirdb.org/miRDB/ accessed on 3 November 2021), and Funrich (http://www.funrich.org/ accessed on 3 November 2021) with default parameters [36]. For miR–17–5p, 98 target genes were identified (Figure 5A), and the network is shown in Figure 5B. Thereafter, the 98 potential target genes were included in the Database for Annotation, Visualization, and Integrated Discovery (DAVID, https://david.ncifcrf.gov/ accessed on 3 November 2021), a functional annotation enrichment algorithm for large-scale biological datasets was used for pathway analysis [37]. The results showed that the FoxO signaling pathway was the top-ranked (Figure 5C). Figure 5D,E illustrated the miR–193a–5p target genes. Particularly, protein processing in the endoplasmic reticulum signaling pathway was the enriched pathway for the target genes of two miRNA (Figure 5C,F).

## 4. Discussion

It is reported that serum miR–483–3p and miR–29c can be served as new biomarkers for the diagnosis of severe pneumonia in children [38,39]. In severe community-acquired pneumonia (SCAP), miR–181b serves as a diagnostic and prognostic biomarker [40]. All these studies were based on free miRNAs in blood, which are easy to be degraded. MiRNAs in extracellular vesicles (EVs-miRNAs) are encapsulated in a bilayer lipid membrane, which protects them from degrading enzymes, such as ribonucleases, and confers outstanding stability in body fluids. In fact, stability is particularly important when developing novel biomarkers using body fluids [41,42]. For EVs-miRNAs, serum EVs-miRNAs have been viewed as candidate diagnostic biomarkers in children with adenovirus-infected pneumonia [43]. Another study outlined the possibility of using miRNAs in serum extracellular vesicles (EVs) to differentiate patients with CAP from healthy volunteers [7]. Although, circulating EVs-miRNAs in the blood are influenced by systemic pathophysiological states, and their levels and contents are altered in several disease states, including cardiovascular disease [44]. Our study revealed that, in patients with pneumonia, miR–17–5p and miR–193a–5p were remarkably elevated in BALF-EVs but not in plasma-EVs, compared with the control group. This observation implies that miRNAs in plasma-EVs are not effective in identifying pulmonary infections in critical patients on mechanical ventilation with similar baseline characteristics. In contrast, BALF-EVs derived from patients, which can specifically and factually respond to lung pathology, were rarely investigated. The existing studies have established the mouse model of pulmonary inflammation for obtaining BALF-EVs and conducting in vitro experiments on respiratory-related cell lines [25].

In this study, the AUC of PCT, miR–17–5p, and miR–193a–5p in BALF-EVs were 0.685, 0.753, and 0.692, respectively, indicating that miR–17–5p and miR–193a–5p in BALF-EVs have diagnostic value not inferior to that of PCT. Moreover, the two miRNAs had higher specificity and are expected to become biomarkers for the diagnosis of pneumonia. Among the three indicators investigated, miR–17–5p in BALF-EVs provided the highest diagnostic performance (the AUC, 0.753). MiR–193a–5p in BALF-EVs had the highest specificity (100%) but the sensitivity (50.82%) is somewhat lower. It means that patients, whose expression of miR–193a–5p in BALF-EVs is below the cutoff value, are at low risk of pulmonary infection. For these patients, empiric therapy without a confirmed infection can be appropriately reduced to avoid unnecessary antibiotic exposure. This benefit may have a crucial impact on public health, especially for countries with excessive antibiotic consumption.

In addition, we measured the candidate miRNAs in extracellular vesicles from human alveolar epithelial cells (A549). Quite interestingly, these two miRNAs could not be detected. Macrophages play a crucial role in innate immunity and host defense against infectious diseases [45]. BALF-EVs collected from mice with intratracheal LPS instillation were mainly secreted from lung macrophages [25]. Consistently in our study, miR–17–5p and miR–193a–5p were strikingly upregulated in EVs from LPS-treated tMACs, compared with controls. One point to make is that flow cytometry analysis showed the tMACs were differentiated into M1 macrophages by stimulating them with 1 μg/mL of LPS for 24 h (Appendix A). Due to cell phenotype changes, the possibility that differentiated cells release extracellular vesicles with distinct compositions cannot be excluded. Therefore, this article should be considered a preliminary report and the relationship between the change of phenotype of tMACs and the expression of miR–17–5p and miR–193a–5p in EVs should be deeply investigated in the future.

Our results prompt the idea that the extracellular vesicles carrying a high abundance of miR–17–5p and miR–193a–5p in bronchoalveolar lavage fluid may be derived from lung macrophages rather than alveolar epithelial cells. Although previous articles have reported that miR–17–5p was upregulated in EVs from BALF of patients with influenza A virus (IAV) -induced acute respiratory distress syndrome and from the supernatant of IAV-infected lung epithelial cells (A549) [34]. This difference may be attributable to the different intervening factors or different disease models we used. Additionally, we revealed that the candidate miRNAs levels in EVs derived from macrophage is dynamic in inflammatory response; characteristically, they first rise and gradually recover. The results indicate that miR–17–5p and miR–193a–5p in extracellular vesicles have potential value in assisting early recognition of inflammatory responses.

Previously, miR–17–5p and miR–193a–5p were both identified as candidates for targeted therapy in many diseases, particularly in cancer therapy [46,47,48,49,50,51]. To further explore the molecular mechanisms of miR–17–5p and miR–193a–5p in infective lung disease, bioinformatics analyses were performed to identify the miRNAs target genes and significant pathways of these target genes. The target genes of the miR–17–5p are significantly enriched in the forkhead box O (FoxO) signaling pathway, which is involved in many cellular physiological events [52]. FoxO transcription factor is involved in the exudative phase of acute lung injury by regulating endothelial stromelysin1 [53]. FoxO transcription factor regulates the cigarette smoke extract-induced autophagy of alveolar epithelial cell A549 [54]. The signaling pathway that drew our attention was protein processing in the endoplasmic reticulum, which plays a vital role in the control of the progression of the cell cycle, differentiation, inflammation, aging, and immunity [55,56,57]. Whether the miR–17–5p and miR–193a–5p are involved in the above pathway needs to be further investigated.

Our study had several limitations. First, all patients were recruited from a single center, and the sample size was relatively small. Second, the exact mechanisms of how these miRNAs function in lung inflammation are still unclear. Future studies are needed to confirm the actual regulatory targets and biological functions of the discovered miRNAs to obtain practical experimental evidence of the mechanistic processes involved in pneumonia.

In conclusion, our study indicated that, compared with non-respiratory infection patients, miR–17–5p and miR–193a–5p in extracellular vesicles derived from bronchoalveolar lavage fluid were significantly increased in the patients with pulmonary infection. MiR–17–5p and miR–193a–5p in extracellular vesicles derived from bronchoalveolar lavage fluid may be used as a new biomarker for the diagnosis of pneumonia.

## Figures and Tables

**Figure 1 cells-11-02961-f001:**
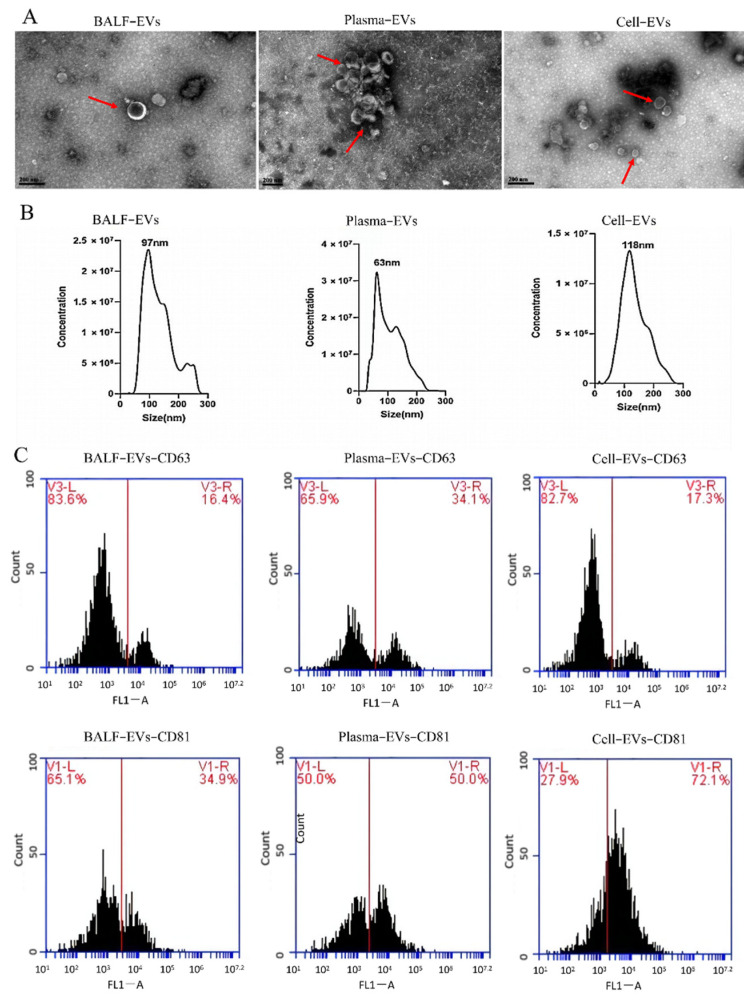
Characterization of extracellular vesicles (EVs) derived from different samples. (**A**) Transmission electron microscopy (TEM) images of EVs derived from bronchoalveolar lavage fluid (BALF-EVs), plasma (plasma-EVs), and cell culture supernatant (cell-EVs). Scale bars, 200 nm. (**B**) Different samples’ EVs particles were investigated by nanoparticle tracking analysis (NTA). (**C**) Markers for EVs (CD63 and CD81) were detected by flow cytometry analysis.

**Figure 2 cells-11-02961-f002:**
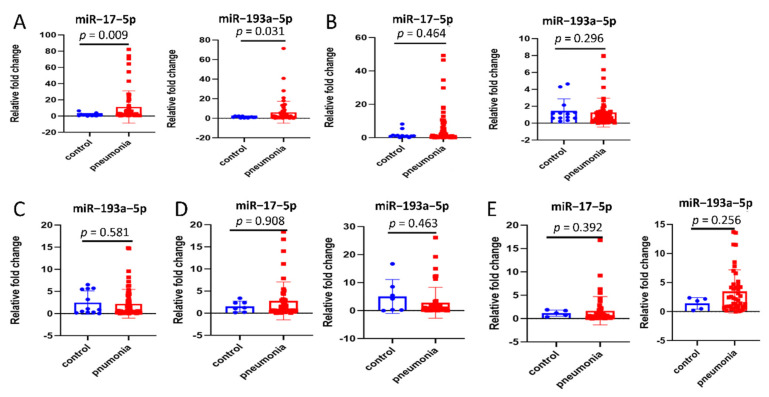
Differentially expressed miRNAs specifically shown in extracellular vesicles derived from bronchoalveolar lavage fluid (BALF-EVs). (**A**) Comparison of miR–17–5p and miR–193a–5p in BALF-EVs between control group (n = 13) and pneumonia group (n = 61). (**B**) Comparison of miR–17–5p and miR–193a–5p in cell-debris pellets derived from BALF (BALF-cell-debris pellets) between control group (n = 13) and pneumonia group (n = 61). (**C**) Comparison of miR–193a–5p in EVs-free supernatants derived from BALF (BALF-EVs-free supernatants) between control group (n = 13) and pneumonia group (n = 61). (**D**) Comparison of miR–17–5p and miR–193a–5p in EVs derived from plasma (plasma-EVs) between control group (n = 7) and pneumonia group (n = 48). (**E**) Comparison of miR–17–5p and miR–193a–5p in total plasma between control group (n = 5) and pneumonia group (n = 44). Data presented as a relative fold change for each miRNA. Box plots are displayed, where the horizontal bar represents the median, the box represents the IQR, and the whiskers represent the maximum and minimum values. Comparisons were made using the Mann–Whitney U test. IQR, interquartile range.

**Figure 3 cells-11-02961-f003:**
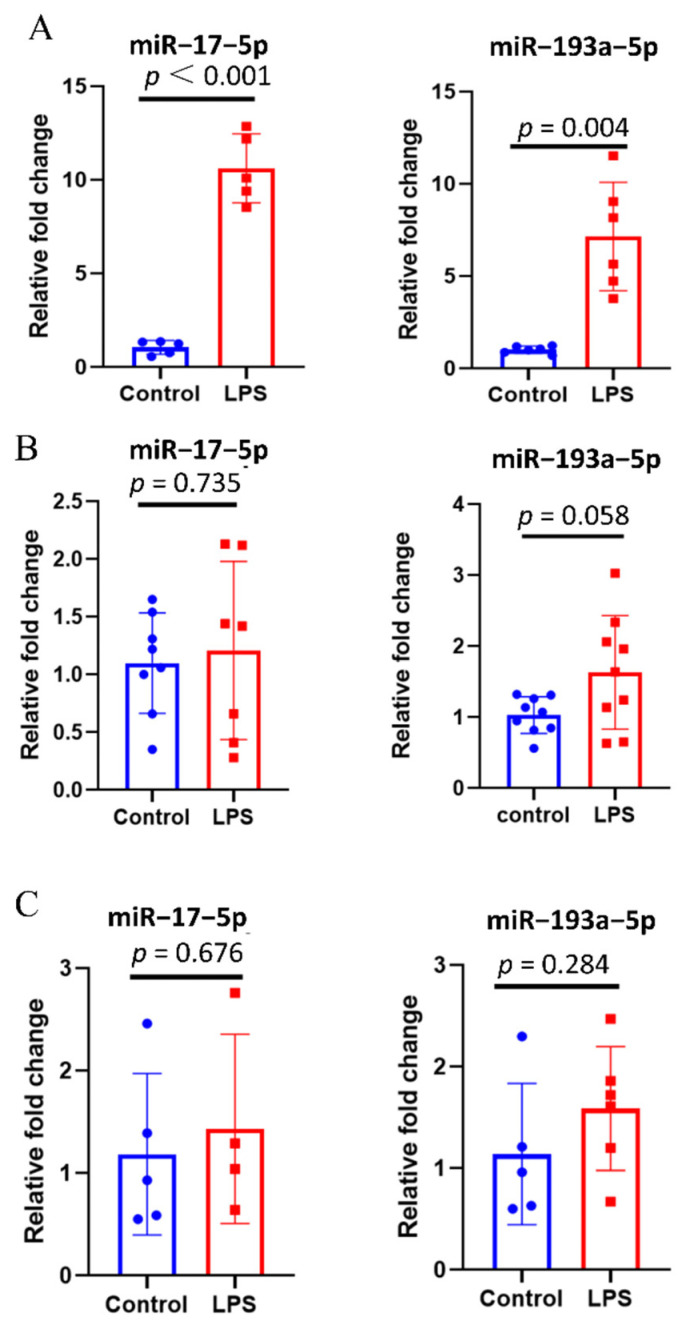
Inflammatory stimuli increase the expression of miR–17–5p and miR–193a–5p in EVs derived from macrophages. (**A**) Comparison of the expression of miR–17–5p and miR–193a–5p in EVs from THP-1-derived macrophages (tMACs-EVs, n = 5–6) between two groups. (**B**) Comparison of the expression of miR–17–5p and miR–193a–5p in tMACs (n = 7–9) between two groups. (**C**) Comparison of the expression of miR–17–5p and miR–193a–5p in EVs-free supernatants derived from tMACs (tMACs-EVs-free supernatants, n = 4–6) between two groups. Data presented as a relative fold change for each miRNA. Box plots are displayed, where the horizontal bar represents the median, the box represents the IQR, and the whiskers represent the maximum and minimum values. Comparisons were made using the unpaired t test.

**Figure 4 cells-11-02961-f004:**
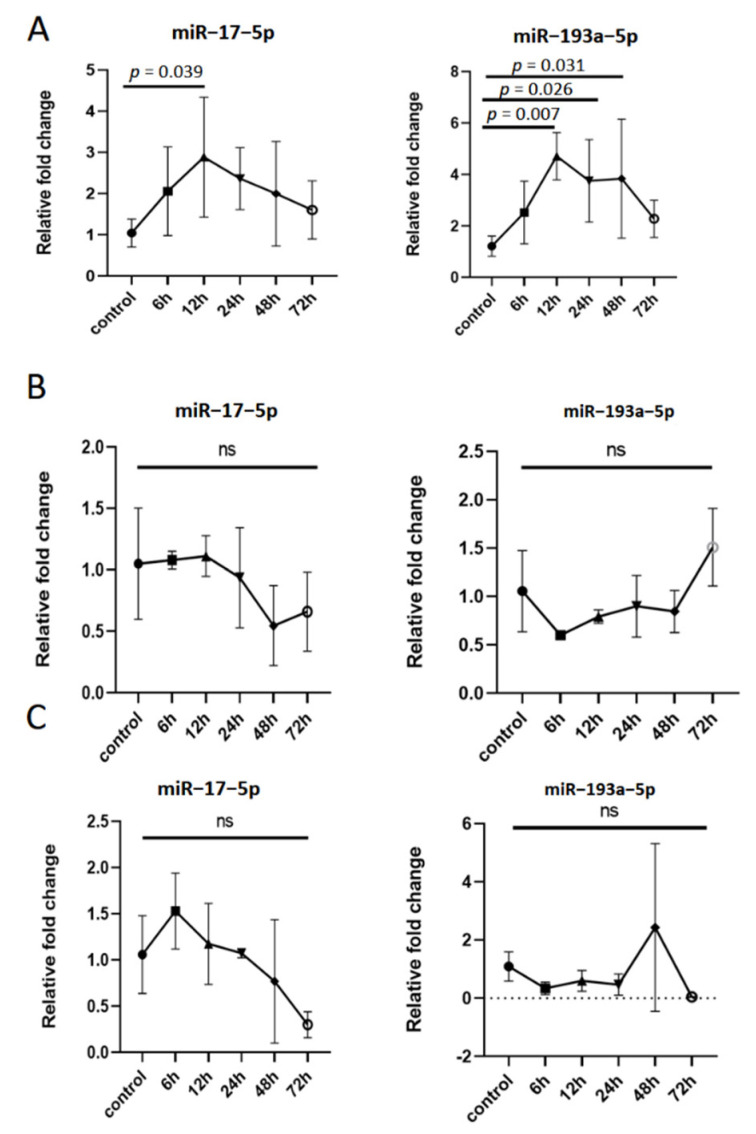
The expression of miR–17–5p and miR–193a–5p in extracellular vesicles derived from macrophages is dynamic in the inflammatory response. tMACs cells were treated with PBS (control) or LPS (1 μg/mL) for 6, 12, 24, 48, and 72 h. (**A**) Under constant inflammatory stimulation, the time and concentration curves of miR–17–5p and miR–193a–5p in extracellular vesicles derived from tMACs (tMACs-EVs, n = 6). (**B**) Under constant inflammatory stimulation, the time and concentration curves of miR–17–5p and miR–193a–5p in tMACs (n = 4). (**C**) Under constant inflammatory stimulation, the time and concentration curves of miR–17–5p and miR–193a–5p in EVs-free supernatants derived from tMAC (tMAC-EVs-free supernatants, n = 4). ns: not statistical.

**Figure 5 cells-11-02961-f005:**
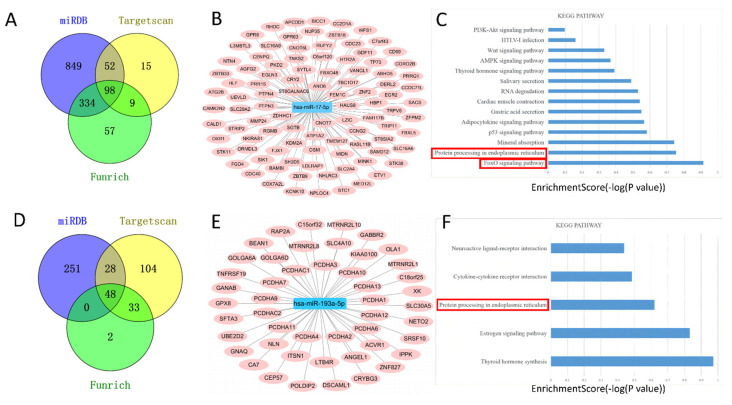
MiRNAs target prediction and pathway analysis. TargetScan, Funrich, and miRDB were used to predict downstream target genes of miR–17–5p (**A**) and miR–193a–5p (**D**). Construction of miR–17–5p (**B**) and miR–193a–5p (**E**) centered target gene regulatory network. KEGG analysis of the enrichment pathway of miR–17–5p target genes (**C**) and miR–193a–5p target genes (**F**).

**Table 1 cells-11-02961-t001:** Demographic and baseline characteristics of patients.

	Control Group (n = 13)	Pneumonia Group (n = 61)	*p* Value
Age, years, mean ± SD	50.38 ± 13.16	57.70 ± 18.84	0.107
Sex			1.000
Male	11	49	
Female	2	12	
Any comorbidity, n (%)	12 (92.3)	53 (0.87)	0.940
Hypertension	9 (69.2)	26 (42.6)	0.081
Diabetes	3 (23.1)	14 (23.0)	1.000
Malignancy	0 (0.0)	5 (8.2)	0.579
Chronic liver disease	2 (15.4)	14 (23)	0.818
Cardiovascular disease	3 (23.1)	10 (10.6)	0.862
Renal disease	1 (7.7)	9 (14.8)	0.819
Craniocerebral trauma	9 (69.2)	18 (29.5)	0.017
hsCRP, mean (range), (mg/L)	52.4 (0.9–151.7)	76.5 (1.7–328.7)	0.236
PCT, mean (range), (ng/mL)	1.1 (0.1–4.1)	4.9 (0.02–45.0)	0.004

Abbreviations: SD, standard deviation; hsCRP, high-sensitivity C-reactive protein; PCT, procalcitonin.

**Table 2 cells-11-02961-t002:** Diagnostic value of each index of pneumonia in critical patients.

Index	Cutoff Value	AUC	95% CI	Specificity (%)	Sensitivity (%)	*p*
miR–17–5p	2.32	0.753	0.639–0.846	84.62	59.02	0.0002
miR–193a–5p	2.40	0.692	0.574–0.794	100	50.82	0.0028
miR–17–5p + miR–193a–5p	−	0.748	0.633–0.842	92.31	57.38	0.0001
hsCRP	28.80 mg/L	0.651	0.531–0.758	61.54	80.33	0.1420
PCT	0.89 ng/mL	0.685	0.566–0.788	69.23	62.30	0.0254

Abbreviations: hsCRP, high-sensitivity C-reactive protein; PCT, procalcitonin; AUC, area under the curve; CI, confidence interval.

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
