# Peer review of "Diagnostic Potential of microRNAs in Extracellular Vesicles Derived from Bronchoalveolar Lavage Fluid for Pneumonia—A Preliminary Report"

_cells, 2022, doi:10.3390/cells11192961_

Round 1

Reviewer 1 Report

The authors focused on the potential diagnostic role for exo- microRNAs for pneumonia patients and the possibility to use them like new biomarker for a rapid and early diagnosis.Basically the experimental design appear clear and proper for your goal despite as you know the small size of your cohort.

Here there are some suggestions to improve the manuscript:

·         English should be revised because of some mistakes (i.e. EX0s in 3.3 paragraph title)

·         add an abbreviation list or specify all abbreviations used in this work

·         in the discussion paragraph you explain what are microRNA and the knowledge about this topic, maybe could be better to move this part at the begin of this paragraph or in the abstract

·         revise table 1 to allow the people to understand clearly and immediately all the information described there

·         pay attention about your graph: in fig. 3 

·         in 3.4 paragraph you talk about PBS group: “compared with the PBS group… “what is PBS group? Is it your control group? Please specify that information in the text, not only for the figure

·         In Figure 4A, the authors describe the expression of macrophages-exosomal miR-17-5p and miR-193a-5p in in the inflammatory response. As described in the figure, the t-MACs were treated with LPS; however, there is a discrepancy. The materials and methods describe the treatment with LPS for 24h, in figure 4 a treatment with a timing ranging from 6h to 72h is represented. For what reason? The authors should explain this choice.

·         3.7 paragraph describe the potential target genes of miR-17-5p and miR-193a-5p maybe could be interesting evaluate by western blot analysis some protein of FoxO pathway and some protein involved in the endoplasmic reticulum signaling that seem the main pathways affected by these microRNA

Reviewer 2 Report

General comments:

I)      Materials and methods lacked essential information to assess the reproducibility of the experiments.

II)    It was unclear how the phenotypes of the cells were evaluated and if they were affected during the LPS treatment.

III)   It was unclear how many independent experiments were performed to validate the significance of the statistics. 

IV)  Since, the overall distribution indicated a non-negligible population of medium-sized vesicles, replace “exosomes” by “small-sized vesicles and medium-sized vesicles”.

V)    It was unclear if argonautes as microRNAS binding proteins were associated to the extracellular vesicles. Further experiments are need to consider this possibility.

VI)  As pointed by the authors, the work had limitations: in lines 350-352, p11 “Due to the the overall small sample size and the heterogeneity of the patient population, a larger randomized study is required to confirm the findings from this study. ”;   Lines  383-385, p12 “Whether the exosomal miR-17-5p and miR-193a-5p are involved in the above pathway needs to be further investigated”; Lines 386-391 “Our study had several limitations. First, all patients were recruited from a single center, and the sample size was relatively small. Second, exact mechanisms of how these miRNAs function in lung inflammation are still unclear. Future studies are needed to confirm the actual regulatory targets and biological functions of the discovered miRNAs to obtain practical experimental evidence of the mechanistic processes involved in pneumonia. ”; Lines 396-398, p11 “To achieve the successful use of these miRNAs as biomarkers, studies in larger patient cohorts will be required to confirm existing results.”

VII) The discussion contained several pargrapgs which could be merged into the introduction. 

Minor comments:

1)    Abstract, lines 25- 26, p1: Delete “Early and appropriate antibiotic treatment reduces morbidity and mortality of pneumonia.However, early identification of pulmonary infection remains a challenge for clinicians.”

2)    Abstract, lines 34-35, p1: Rephrase “Interestingly, miR-17-5p and miR-193a-5p in BALF-cell-debris pellets, BALF-EXO-free supernatants, total plasma, and plasma-EXOs did not differ significantly between both groups.”

3)    Abstract, lines 35-36, p1: Specify the abbreviation LPS in “In vitro experiments revealed that miR-17-5p and miR-193a-5p were strikingly upregulated in exosomes  derived from macrophages stimulated by LPS. 

4)    Abstract, lines 41-43, p1: Specify the abbreviation FoxO  in “Exosomal miR-17-5p and miR-193a-5p may be involved in lung inflammation by influencing FoxO signaling pathway and protein processing  in endoplasmic reticulum.”

5)    Abstract, lines 43-44, p1: It is too broad, rephrase “This study is one of the few studies on BALF-EXO-miRNAs in patients 43 with pneumonia, providing potential diagnostic biomarkers and therapeutic targets for pneumonia.”

6)    Introduction, lines 49-50, p2: Add references to support “According to WHO, Lower respiratory tract infections (LRTIs) are the fourth leading  cause of death globally, causing 2.6 million deaths worldwide in 2019.”

7)    Introduction, lines 52-55, p2: Add references to support “Due to their critical condition, immunodeficiency and requirement of some invasive medical means such as tracheal intubation, most patients in the intensive care unit (ICU) are prone to pulmonary infection.”

8)    Introduction, lines 60-63, p2: Report diagnosis methods, their weaknesses and then add “Therefore, early identification of pulmonary infection, carrying out pathogenic microbiological diagnosis and timely and effective anti-infection measures are of great significance to improve the prognosis of patients

9)    Introduction, lines 66-68, p2: add references to support “However, abnormal clinical manifestations and pulmonary imaging are delayed than infection and failed to identify subclinical disease states. 

10) Introduction, lines 68-70, p2: add references to support “Therefore, multiple biomarkers are used to assist in early identification of pulmonary infection, C-reactive protein (CRP) and procalcitonin (PCT) are the most commonly used.”

11) Introduction, lines 73-74, p2: add references to support “ Therefore, the low specificity of CRP limits its application value as a biomarker for 73 early specific identification of pulmonary infection.”

12) Introduction, lines 94-98, p2: Justify, and rephrase by splitting in several sentences since it is too long “Therefore, from the perspective of exosomes derived from clinical samples (bronchoalveolar lavage fluid and plasma), this study explored the differential expression of key exosomal miRNAs in critical patients with pneumonia and evaluated their auxiliary diagnostic value, so as to explore new specific biomarkers to assist in the diagnosis of pulmonary infection.”

13) Materials and methods, 2.1  Patients and Samples, lines 113-114, p3: Add more information to support  “This study was approved by the Ethics Committee of The Third Affiliated 113 Hospital of Sun Yat-sen University (approval no. [2020] 02-254-02).”

14) Materials and methods, 2.1  Patients and Samples, lines 115-117, p3: Add more information, references and specify BALF abbreviation in “The pulmonary and subpulmonary segments of the selected patients were lavaged 115 with saline using electronic fiber bronchoscopy, and 15 mL BALF was collected (to avoid 116 oral contamination) in sterile sputum collection tubes”

15) Materials and Methods, 2.2. Cell culture and treatment, lines 121-123, p3: Add explanations and justify the use of THP-1, and A549 cells to support “Human monocytes, THP-1, and human lung adenocarcinoma cell line, A549 cells, were maintained in RPIM1640 and DMEM medium, respectively, containing 10% heat-inactivated fetal bovine serum (FBS), 1% penicillin and streptomycin at 37°C, and 5% CO2.”

16) Materials and Methods, 2.2. Cell culture and treatment, lines 121-123, p3: It was unclear if cell phenotypes were checked and how it was performed to support “THP-1 cells were cultured in maintenance media and supplemented with 100 ng/mL 124 phorbol 12-myristate 13-acetate (PMA) for 24 hours to differentiate them into macro- 125 phages (THP-1 derived macrophages [tMACs]). For the control group, normal cultured 126 A549 cells or tMACs were used; for the lipopolysaccharide (LPS) group, conventionally 127 cultured cells were treated with LPS (1 ug/mL) for 24 hours.”

17) Materials and Methods, 2.3. Exosomal purification and identification, lines 142-144, p3: It was unclear if the PBS buffer had any osmotic effect and if the frozen vesicles were broken in “The resultant 142 pellet (EXOs), which were resuspended with phosphate-buffered saline (PBS) and EXO- 143 free supernatants, was stored at −80°C.”

18) Materials and Methods, 2.3. Exosomal purification and identification, lines 147-149, p3: Specify NTA abbreviation in “Thereafter, exosomal particles were analyzed using NTA (NanoSight NS300, Malvern Instruments, United Kingdom).”

19) Materials and Methods, 2.3. Exosomal purification and identification, from line 150, p3 to line 152 p4: Add more information and references to support “In addition, exosomal positive markers (CD63 and CD81) were detected using BD accuri C6 151 flow cytometer.”

20) Materials and Methods, 2.4. Candidate miRNA selection, title, line 154, p4: Change “miRNA” by “microRNA “to be consistent in all the manuscript since key word “microRNA” was used in the medline search. 

21) Materials and Methods, 2.5. RNA extraction and quantitative reverse transcription PCR (qRT-PCR), line 162, p4: Delete (qRT-PCR) in the Title.

22) Materials and Methods, 2.5. RNA extraction and quantitative reverse transcription PCR (qRT-PCR), lines 163, p4: Add more information to support “Total RNA was harvested using TRIzol, according to the manufacturer’s instructions. 

23) Materials and Methods, 2.5. RNA extraction and quantitative reverse transcription PCR (qRT-PCR), lines 164-165, p4: Add more information to support “The expression of target miRNA was determined using the SYBR Green Master Mix kit  (Takara, Japan).

24) Materials and Methods, 2.6. Statistical analysis, lines 169-171, p4: Specify number of independent measurements to support “The continuous variables are expressed as mean ± SD values (normally distributed), or as medians [interquartile ranges] (non-normally distributed), and were analyzed using the Student’s t-test or Mann–Whitney U test, respectively.”

25) Results, 3.2. Characterization of EXOs derived from different samples, title, line 194, p5: Replace EXOs by small-sized and medium-sized vesicles in the title.

26) Results, 3.1. Patient characteristics, lines 186-188, p4:  Specify abbreviation PCT in “The inflammation indicator, PCT, was significantly higher in the pneumonia group than in the control group.”

27) Results, 3.2. Characterization of EXOs derived from different samples, lines 195-196 p5: It appears from the Fig 1B that there are no negligible amount of larger-sized vesicles. Replace “exosomes” by “small-sized and medium sized vesicles” in “Transmission electron microscopy (TEM) was used to assess the morphology of exosomes.”

28) Results, 3.2. Characterization of EXOs derived from different samples, lines 195-200, p5: Due to the use of PBS buffer and the frozen conditions to store extracellular vesicles, it was unclear if the vesicles retained their native morphologies. Further experiments are needed to check that the vesicles are well preserved to support “As shown in Figure 1A, exosomes derived from BALF (BALF-EXOs), plasma 1 (plasma-EXOs) and cell culture supernatant (cell-EXOs) have the characteristic cup- shaped morphology. The size distribution profile of three kinds of samples was investigated by nanoparticle tracking analysis (NTA), which revealed a peak”

29) Results, 3.2. Characterization of EXOs derived from different samples, lines 195-198, p5: My computer did not show Figure 1A with sufficient resolution. Check the resolution of Figure 1A to support “As shown in Figure 1A, exosomes derived from BALF (BALF-EXOs), plasma 1 (plasma-EXOs) and cell culture supernatant (cell-EXOs) have the characteristic cup- shaped morphology.”

30) Results, 3.2. Characterization of EXOs derived from different samples, lines 198-200, p5: I cann’t read the number in the abscises of Fig 1B. Enlarge size of numbers in Fig 1B to support “The size distribution profile of three kinds of samples was investi-gated by nanoparticle tracking analysis (NTA), which revealed a peak size of 97nm,63nm and 118nm respectively (Figure 1B).”

31) Results, 3.2. Characterization of EXOs derived from different samples, lines 200-202, p5: I cann’t read the number in the abscises of Fig 1c. Enlarge size of numbers in Fig 1B to support “Furthermore, the presence of exosomal markers, CD63 and CD81, in all vesicles were revealed by flow cytometry (Figure 1C).”

32) Results, 3.3. Differentially expressed miRNAs specifically showed in BALF-EX0s, line 209, p6: Replace BALF-EXOs by BALF small-sized and medium-sized vesicles in the title.

33) Results, 3.3. Differentially expressed miRNAs specifically showed in BALF-EX0s, Title, line 209 p6: Abbreviation shall be consistent in al the text, sometimes miR, microRNA, miRNA were used. Select one abbreviation. 

34) Results, 3.3. Differentially expressed miRNAs specifically showed in BALF-EX0s, line 210-214, p6: Apparently the selection of miRNAS resulted from a literature search, add references associated to each miRNA to support “Eighteen miRNAs (miR-542-3p, miR-16-5p, miR-20a-3p, miR-27a-5p, miR-92a-3p, miR-342-3p, miR-422a, miR-423-5p, miR-582-3p, miR-885-5p, miR-193b-5p, miR-432-5p, miR-493-3p, miR-452-5p, miR-200b-3p, miR-34a-3p, miR-17-5p, and miR-193a-5p), which were previously revealed to have dysregulated expression profiles in inflammatory diseases using RNA sequencing of population sample, were selected.”

35) Results, 3.3. Differentially expressed miRNAs specifically showed in BALF-EX0s, lines 214-217, p6: Select one abbreviation for micro-RNA in all the text. It was unclear if MiR-17-5p and miR-193a-5p originated from argonaute associated miRs or from extracellular vesicles. Further experiments are needed to confirm that miRs originated from vesicles to support “Among them, 16 miRNAs between two groups showed no statistical difference (Supplementary Figure S1). MiR-17-5p and miR-193a-5p levels were remarkably upregulated in BALF-EXOs of the pneumonia group, compared with the control group (Figure 2A).”

36) Results, 3.3. Differentially expressed miRNAs specifically showed in BALF-EX0s, from lines 214, p6 to line 226, p7: Indicate number of independent samples in the figure -2 legend to support the statistical singificances in “Among them, 16 miRNAs between two groups showed no statistical difference (Supplementary Figure S1). MiR-17-5p and miR-193a-5p levels were remarkably upregulated in BALF-EXOs of the pneumonia group, compared with the control group (Figure 2A). Thereafter, the expression of these two miRNAs in BALF-cell-debris pellets (Figure 2B) and BALF-EXOs-free supernatants (Figure 2C) were assessed. However, no significant differences were observed between both groups. Notably, the expression of miR-17-5p in BALF-EXOs-free supernatants was too low to be detected. Additionally, the levels of these two miRNAs in total plasma, plasma-EXOs, and plasma-EXOs-free supernatants were measured. Surprisingly, in plasma-EXOs (Figure 2D), both miR-17-5p and miR-193a-5p showed no difference in level between both groups; the same trend was observed for total plasma (Figure 2E).”

37) Results, 3.3. Differentially expressed miRNAs specifically showed in BALF-EX0s, from lines 223-226, p7: It was unclear what was the difference between Fig 2A (BALF-EXOs), Fig 2D (plasma-EXOs free supernatant) and Fig 2E  (total plasma). Add explanations to support “Surprisingly, in plasma-EXOs (Figure 2D), both miR-17-5p and miR-193a-5p showed no difference in level between both groups; the same trend was observed for total plasma (Figure 2E).

38) Results, 3.3. Differentially expressed miRNAs specifically showed in BALF-EX0s, Figure-2 legend, lines 231-234, p7: Indicate number of independent measurements to support “Box plots are displayed, where the horizontal bar represents the median, the box represents the IQR, and the whiskers represent the maximum and minimum values. Comparisons were made using the Mann–Whitney U test. miRNA, microRNA; IQR, interquartile range.”

39) Results, 3.4. Inflammatory stimuli increases the expression of miR-17-5p and miR-193a-5p in exosomes derived from macrophages, Title, lines 236-237, p7: Replace “exosomes” by small-sized, medium-sized vesicles and argonautes in the Title.

40) Results, 3.4. Inflammatory stimuli increases the expression of miR-17-5p and miR-193a-5p in exosomes derived from macrophages, lines 236-237 ,p7: Check the abbreviation PBS, which was used as “Phosphate buffer saline” and rephrase “Compared with the PBS group, miR-17-5p and miR-193a-5p were strikingly upregulated in EXOs from THP-1-derived macrophages (tMAC-EXOs) of the LPS group, (Figure 3A).”

41) Results, 3.4. Inflammatory stimuli increases the expression of miR-17-5p and miR-193a-5p in exosomes derived from macrophages, Title, lines 242-245, p7: Check the abbreviation PBS, which was used as “Phosphate buffer saline” and rephrase “Compared with the PBS group, miR-17-5p and miR-193a-5p were strikingly upregulated in EXOs from THP-1-derived macrophages (tMAC-EXOs) of the LPS group, (Figure 3A).”

42) Results, 3.4. Inflammatory stimuli increases the expression of miR-17-5p and miR-193a-5p in exosomes derived from macrophages, Title, lines 242-245, p7: Add explanations and references for the use of THP-1 derived macrophages in “Phosphate buffer saline” and rephrase “Compared with the PBS group, miR-17-5p and miR-193a-5p were strikingly upregulated in EXOs from THP-1-derived macrophages (tMAC-EXOs) of the LPS group, (Figure 3A).”

43) Results, 3.4. Inflammatory stimuli increases the expression of miR-17-5p and miR-193a-5p in exosomes derived from macrophages, lines 242-245, p7: It was unclear how the phenotype of THP-1 derived macrophages was monitored and whether it was affected during LPS incubation to support in “Phosphate buffer saline” and rephrase “Compared with the PBS group, miR-17-5p and miR-193a-5p were strikingly upregulated in EXOs from THP-1-derived macrophages (tMAC-EXOs) of the LPS group, (Figure 3A).”

44) Results, 3.4. Inflammatory stimuli increases the expression of miR-17-5p and miR-193a-5p in exosomes derived from macrophages, lines 242-245, p7: It was unclear if miRNAs were associated to argonautes or to vesicles to support “Phosphate buffer saline” and rephrase “Compared with the PBS group, miR-17-5p and miR-193a-5p were strikingly upregulated in EXOs from THP-1-derived macrophages (tMAC-EXOs) of the LPS group, (Figure 3A).”

45) Results, 3.4. Inflammatory stimuli increases the expression of miR-17-5p and miR-193a-5p in exosomes derived from macrophages, Fig-3 legend, line 255, p8: Indicate number of independent measurements to support “Comparisons made using the unpaired t test.”

46) Results, 3.5. The expression of macrophages-exosomal miR-17-5p and miR-193a-5p is dynamic in inflammatory response, Title, lines 257-258, p8: Replace “exosomal” by “vesicles or argonautes” in the title.

47) Results, 3.5. The expression of macrophages-exosomal miR-17-5p and miR-193a-5p is dynamic in inflammatory response, lines 259-266, p8: It was unclear how the phenotype of the cells were monitored and if they were affected during LPS treatment to support  “To explore the relationship between the expression of miR-17-5p and miR-193a-5p in LPS-induced tMAC-EXOs and inflammatory response, the levels of this two miRNAs in LPS-induced tMAC-EXOs were dynamically measured. In tMAC-EXOs (Figure 4A), the expression of miR-193a-5p was drastically increased in a time-dependent manner after LPS treatment and peaked at 12 hours; it gradually returned to normal levels. Concurrently, miR-17-5p showed the same trend. In tMAC-cell-debris pellets and tMAC-EXO- free supernatants, the levels of the candidate miRNAs were unaffected by LPS stimuli (Figure 4B-C).”

48) Results, 3.5. The expression of macrophages-exosomal miR-17-5p and miR-193a-5p is dynamic in inflammatory response, lines 259-266, p8: It was unclear if miRNAs were associated to argonautes or to vesicles in “To explore the relationship between the expression of miR-17-5p and miR-193a-5p in LPS-induced tMAC-EXOs and inflammatory response, the levels of this two miRNAs in LPS-induced tMAC-EXOs were dynamically measured. In tMAC-EXOs (Figure 4A), the expression of miR-193a-5p was drastically increased in a time-dependent manner after LPS treatment and peaked at 12 hours; it gradually returned to normal levels. Concurently, miR-17-5p showed the same trend. In tMAC-cell-debris pellets and tMAC-EXO- free supernatants, the levels of the candidate miRNAs were unaffected by LPS stimuli (Figure 4B-C).”

49) Results, 3.5. The expression of macrophages-exosomal miR-17-5p and miR-193a-5p is dynamic in inflammatory response, Fig-4 Legend, line 275, p9: Indicate number of independent measurements to support “Data were expressed as mean ± SD.”

50) Results, 3.6. Diagnostic value of BALF-exosomal miR-17-5p and miR-193a-5p in critical patients with pneumonia, Title, lines 277-278, p9: Replace “exosomal” by “small-sized vesicles, medium-sized vesicles and argonates” in the title.

51) Results, 3.6. Diagnostic value of BALF-exosomal miR-17-5p and miR-193a-5p in critical patients with pneumonia, lines 281-282, p9: Replace “exosomal” by “small-sized vesicles, medium-sized vesicles and argonates” in “As shown in Table 2, hsCRP had no diagnostic utility. PCT, BALF-exosomal miR-17-5p and miR-193a-5p had acceptable diagnostic value.”

52) Results, 3.6. Diagnostic value of BALF-exosomal miR-17-5p and miR-193a-5p in critical patients with pneumonia, lines 281-282, p9: Indicate number of independent measurements to support in “As shown in Table 2, hsCRP had no diagnostic utility. PCT, BALF-exosomal miR-17-5p and miR-193a-5p had acceptable diagnostic value.”

53) Results, 3.7. MiRNA target prediction and pathway Analysis, lines 292-295, p10: Add explanations and references to support “The roles of miR-17-5p and miR-193a-5p were investigated using bioinformatics. Multiple databases (TargetScan, miRDB, and Funrich) were combined to predict and screen their target genes. For miR-17-5p, 98 target genes were identified (Figure 6A), and the network is shown in Figure 6B.”

54) Results, 3.7. MiRNA target prediction and pathway Analysis, lines 295-296, p10: Add explanations and references to support “Thereafter, the 98 potential target genes were included in the DAVID database for KEGG_PATHWAY analysis.”

55) Results, 3.7. MiRNA target prediction and pathway Analysis, Figure 6, lines 294-299, p10: Enlarge size of letters in the Figure-6 Text to support “For miR-17-5p, 98 target genes were identified (Figure 6A), and the network is shown in Figure 6B. Thereafter, the 98 potential target genes were included in the DAVID database for KEGG_PATHWAY analysis. The results showed that the FoxO signaling pathway was the top-ranked (Figure 6C). Figure 6D-E illustrated the miR-193a5p target genes. Particularly, protein processing in the endoplasmic reticulum signaling pathway was the enriched pathway for the target genes of two miRNA (Figure 6C, F).”

56) Discussion, lines 308-310: Move into the introduction “BALF, including alveolar biochemical components and cells, is obtained by infusing physiological saline into alveoli by bronchoscopy, and then aspirating under negative pressure[19].”

57) Discussion, lines 310-311, p10: Add references, and move into the introduction “Information gained from BALF is regarded to be a complement of lung biopsy pathology.”

58) Discussion, lines 311-314, p10: Move into the introduction “In addition, lavage samples a much larger area of the lungs than can be obtained by the small tissue fragments of transbronchial biopsy or by open biopsy specimens, therefore giving a more representative picture of inflammatory and immunologic changes[20].”

59) Discussion, lines 314-317, p10: Replace “exosomes” by “extracellular vesicles” since there were a mixture of small-sized and medium-sized vesicles and move into the introduction  “Exosomes, as an important component of BALF, have been proved to be a new intercellular communication medium and play an important role in the occurrence and development of various pulmonary diseases, such as asthma, chronic obstructive pulmonary disease and lung cancer[21-23].”

60) Discussion, lines 317-318, p10: It was already stated in the introduction, delete “However, few studies have focused on the clinical application value of miRNAs derived from exosomes in bronchoalveolar lavage fluid of pneumonia.”

61) Discussion, lines 324-328 , p11: Replace “exosomal” by “extracellular vesicles” in “Exosomal miRNAs are encapsulated in a bilayer lipid membrane, which protects them from degrading enzymes, such as ribonucleases, and confers an outstanding stability in body fluids. In fact, the stability is particularly important when developing novel biomarkers using body fluids [27, 28]. For exosomal miRNAs, serum exosomal miRNAs have been viewed as candidate diagnostic biomarkers in children with adenovirus-infected pneumonia [29].”

62) Discussion, lines 331-332, p11: Add references to support “Whereas circulating blood exosomal microRNAs are influenced by systemic pathophysiological states.” 

63) Discussion, lines 340-342, p11: Replace “exosomal” by “extracellular vesicles and argonautes” and rephrase to support  “In this study, the AUC of PCTBALF-exosomal miR-17-5p and miR-193a-5p were 0.685, 0.753 and 0.692, respectively, indicating that BALF-exosomal miR-17-5p and miR-193a-5p have diagnostic value not inferior to that of PCT.”

64) Discussion, lines 344-348, p11: Replace “exosomal” by “extracellular vesicles and argonautes” and rephrase to support  “Among the three indicators investigated, BALF-exosomal miR-17-5p provided the highest diagnostic performance (the AUC, 0.753). BALF-exosomal miR-193a-5p  had the highest specificity (100%) but the specificity (50.82%) is somewhat lower. It means that patients with BALF-exosomal miR-193a-5p levels below the cutoff are at low risk of pulmonary infection.”

65) Discussion, lines 348-349, p11: Rephrase “For these patients, empirical antibiotic use can be appropriately reduced to avoid unnecessary antibiotic exposure.”

66) Discussion, lines 353-357, p11: Move into the introduction “BALF-EXOs are secreted by various cell types including epithelial cells, alveolar macrophages, endothelial cells, wherein alveolar macrophages and epithelial cells are the main sources [30]. Alveolar epithelial cells are parenchymal cells of lung tissue, which can secrete a variety of cytokines to participate in the occurrence and injury of lung tissue inflammation, and are crucial for maintaining the integrity and function of lung.[31, 32].”

67) Discussion, lines 363-365, p11: Replace “exosomes” by “extracellular vesicles and argonautes” and rephrase to support “Our results prompts that the exosomes carrying high abundance of miR-17-5p and miR-193a-5p in bronchoalveolar lavage fluid may be derived from lung macrophages rather than alveolar epithelial cells”

68) Discussion, move into the introduction, lines 370-374, p11: “MiRNAs are small non-coding RNAs that have the capability of regulating gene expression by promoting their target messenger RNA (mRNA) degradation or inhibiting the translation of target genes [34, 35]. Previously, miR-17-5p and miR-193a-5p were both identified as candidates for targeted therapy in many diseases, particularly in cancer therapy [36-41].”

69) Discussion, from line 376, p11 to line 378, p12: Add references to support “The target genes of the miR- 376 17–5p are significantly enriched in FoxO signaling pathway, which is involved in many cellular physiological events.”

Reviewer 3 Report

To make their diagnostic tool, why did the authors not add mir-17-5p and mir-193a-5p together for the calculation of the AUC curve, specificity and sensitivity.

The introduction explains that the diagnosis of bacterial pneumonia should be made within 4 hours, if possible to have the best benefit for the patients. The centrifugation steps of the BAL fluid to isolate BAL exosomes takes more than 2 hours already.  How is this test going to work in the clinic.  Or perhaps the introduction should explain the steps better that would take this test towards the clinic.

I think that this article may be of interest for the discussion : Scheller, N. et al. "Proviral microRNAs detected in extracellular vesicles from bronchoalveolar lavage fluid of patients with influenza virus–induced acute respiratory distress syndrome." The Journal of infectious diseases 219.4 (2019): 540-543.

line 346 - sensitivity

Round 2

Reviewer 2 Report

General comments (second round): 

II) It was unclear how the phenotypes of the cells were evaluated and if they were affected during the LPS treatment.

Response: Thanks for your suggestion. According to your suggestion, we have added the related references which used THP-1 derived macrophages as a model cell line. in“2.2. Cell culture and treatment”. As follow:” THP-1 cells were cultured in maintenance media and supplemented with 100 ng/mL phorbol 12-myristate 13-acetate (PMA) for 24 hours to differentiate them into macro-phages (THP-1 derived macrophages [tMACs])[28].”

In our study, we did not check the cell phenotypes of THP-1 derived macrophages [tMACs].It is because that it is widely reported that THP-1 cells are the most commonly used precursors for generating macrophages in vitro, and following differentiation using PMA, THP-1 cells acquire a macrophage-like phenotype[1-3]. In addition, the phenotype of THP-1 derived macrophages was affected during LPS incubation. In this article we mainly focus on the potential diagnostic role for BALF- exo-miR-17-5p and miR-193a-5p for pneumonia. Due to the presence of several interfering factors in clinical samples, in vitro experiments (THP-1 derived macrophages were treated with LPS) were conducted to verify that the inflammatory stimuli increases the expression of exosomal miR-17-5p and miR-193a-5p. And our results prompts that the exosomes carrying high abundance of miR-17-5p and miR-193a-5p in bronchoalveolar lavage fluid may be derived from lung macrophages rather than alveolar epithelial cells. And during LPS incubation, the relationship between the change of phenotype of THP-1 derived macrophages and the expression of exosomal- miR-17-5p and miR-193a-5p will be deeply investigated in our future biological research.

[1].      Murray, P.J., et al. Macrophage activation and polarization: nomenclature and experimental guidelines. Immunity, 2014. 41(1): p. 14-20.

[2].      Maeß, M.B., et al. Reduced PMA enhances the responsiveness of transfected THP-1 macrophages to polarizing stimuli. J Immunol Methods, 2014. 402(1-2): p. 76-81.

[3].      Lund, M.E., et al.The choice of phorbol 12-myristate 13-acetate differentiation protocol influences the response of THP-1 macrophages to a pro-inflammatory stimulus. J Immunol Methods, 2016. 430: p. 64-70.

Reviewer’s answer: The reason why I insisted to check the phenotype of the cell is that LPS incubation modified the phenotype of the cells, confirmed by the author’s answer. Therefore, due to cell phenotype changes, the possibility that differentiated cells release extracellular vesicles with distinct composition can not be excluded. The findings appear preliminary as pointed by the authors “And during LPS incubation, the relationship between the change of phenotype of THP-1 derived macrophages and the expression of exosomal- miR-17-5p and miR-193a-5p will be deeply investigated in our future biological research.” 

V) It was unclear if argonautes as microRNAS binding proteins were associated to the extracellular vesicles. Further experiments are need to consider this possibility.

Response: Thanks for your constructive suggestion. In this article we mainly focus on the potential diagnostic role for exo-miR-17-5p and miR-193a-5p for pneumonia patients and the possibility to use them like new biomarker for a rapid and early diagnosis. This intriguing possibility that if argonautes as microRNAs binding proteins will be deeply investigated in our future biological research.

Reviewer’s answer: The title of the manuscript “Diagnostic potential of microRNAs in extracellular vesicles derived from bronchoalveolar lavage fluid for pneumonia” and its scope claim that extracellular vesicles carry exo-miR-17-5p and miR-193a-5p. However, the possibility of argonautes to carry microRNAs was not checked. The title and the scope of the manuscript shall be corrected. 

VI) As pointed by the authors, the work had limitations: in lines 350-352, p11 “Due to the overall small sample size and the heterogeneity of the patient population, a larger randomized study is required to confirm the findings from this study. ”;   Lines  383-385, p12 “Whether the exosomal miR-17-5p and miR-193a-5p are involved in the above pathway needs to be further investigated”; Lines 386-391 “Our study had several limitations. First, all patients were recruited from a single center, and the sample size was relatively small. Second, exact mechanisms of how these miRNAs function in lung inflammation are still unclear. Future studies are needed to confirm the actual regulatory targets and biological functions of the discovered miRNAs to obtain practical experimental evidence of the mechanistic processes involved in pneumonia. ”; Lines 396-398, p11 “To achieve the successful use of these miRNAs as biomarkers, studies in larger patient cohorts will be required to confirm existing results.”

Response: Thanks for your suggestion. To be honest, our study had several limitations. In the next step, we will perform a larger randomized study to confirm the potential diagnostic role of miR17-5p and miR193a-5p in BALF-EVs for pneumonia and the actual regulatory targets and biological functions of the discovered miRNAs.

Reviewer’s answer: It was not addressed. The manuscript contains significant limitations. 

Reviewer 3 Report

Thank you for  carefully addressing my comments.

Author Response

Thank you very much for your constructive comments, which have greatly improved the quality of our article. Thank you again and wish you all the best.